# The Lack of Synergy between Carvedilol and the Preventive Effect of Dexrazoxane in the Model of Chronic Anthracycline-Induced Cardiomyopathy

**DOI:** 10.3390/ijms241210202

**Published:** 2023-06-15

**Authors:** Jaroslaw Szponar, Erwin Ciechanski, Marta Ostrowska-Lesko, Agnieszka Gorska, Michal Tchorz, Anna Dabrowska, Jaroslaw Dudka, Marek Murias, Michał Kowalczyk, Agnieszka Korga-Plewko, Slawomir Mandziuk

**Affiliations:** 1Toxicology Clinic, Faculty of Medicine, Medical University of Lublin, 100 Krasnik Avenue, 20-550 Lublin, Poland; jaroslaw.szponar@umlub.pl (J.S.); agnieszka.gorska@umlub.pl (A.G.); michal.tchorz@umlub.pl (M.T.); 2Clinical Department of Toxicology and Cardiology, Regional Specialist Hospital, 100 Krasnik Avenue, 20-550 Lublin, Poland; 3Clinical Department of Cardiology, Regional Specialist Hospital, 100 Krasnik Avenue, 20-550 Lublin, Poland; erwin.ciechanski@umlub.pl; 4Chair and Department of Toxicology, Medical University of Lublin, 8b Jaczewski Street, 20-090 Lublin, Poland; anna.dabrowska@umlub.pl (A.D.); jaroslaw.dudka@umlub.pl (J.D.); 5Chair and Department of Toxicology, Poznan University of Medical Sciences, 30 Dojazd Street, 60-572 Poznan, Poland; marek.murias@ump.edu.pl; 6First Department of Anaesthesiology and Intensive Care, Medical University of Lublin, ul. Jaczewskiego 8, 20-954 Lublin, Poland; michalkowalczyk@vp.pl; 7Independent Medical Biology Unit, Medical University of Lublin, 8b Jaczewski Street, 20-090 Lublin, Poland; agnieszkakorgaplewko@umlub.pl; 8Department of Pneumology, Oncology and Allergology, Medical University of Lublin, 8 Jaczewski Street, 20-090 Lublin, Poland; slawomir.mandziuk@umlub.pl

**Keywords:** post-anthracycline cardiotoxicity, cardioprotection, dexrazoxane, carvedilol, doxorubicin, cardiac molecular metabolism

## Abstract

The anticancer efficacy of doxorubicin (DOX) is dose-limited because of cardiomyopathy, the most significant adverse effect. Initially, cardiotoxicity develops clinically silently, but it eventually appears as dilated cardiomyopathy with a very poor prognosis. Dexrazoxane (DEX) is the only FDA-approved drug to prevent the development of anthracycline cardiomyopathy, but its efficacy is insufficient. Carvedilol (CVD) is another product being tested in clinical trials for the same indication. This study’s objective was to evaluate anthracycline cardiotoxicity in rats treated with CVD in combination with DEX. The studies were conducted using male Wistar rats receiving DOX (1.6 mg/kg b.w. *i.p.*, cumulative dose: 16 mg/kg b.w.), DOX and DEX (25 mg/kg b.w. *i.p.*), DOX and CVD (1 mg/kg b.w. *i.p.*), or a combination (DOX + DEX + CVD) for 10 weeks. Afterward, in the 11th and 21st weeks of the study, echocardiography (ECHO) was performed, and the tissues were collected. The addition of CVD to DEX as a cardioprotective factor against DOX had no favorable advantages in terms of functional (ECHO), morphological (microscopic evaluation), and biochemical alterations (cardiac troponin I and brain natriuretic peptide levels), as well as systemic toxicity (mortality and presence of ascites). Moreover, alterations caused by DOX were abolished at the tissue level by DEX; however, when CVD was added, the persistence of DOX-induced unfavorable alterations was observed. The addition of CVD normalized the aberrant expression of the vast majority of indicated genes in the DOX + DEX group. Overall, the results indicate that there is no justification to use a simultaneous treatment of DEX and CVD in DOX-induced cardiotoxicity.

## 1. Introduction

Anthracycline-based (ANT) chemotherapy regimens have been the cornerstone of many cancer treatments for decades [1]. ANTs, on the other hand, can have mild to severe short- and long-term harmful consequences. Anthracycline-induced dilated cardiomyopathy (AIDC) is one of the most severe implications of anthracycline therapy that can develop over time [1,2]. AIDC is caused by cardiac remodeling, which results in a decrease in the left ventricular ejection fraction, which has a very unfavorable prognosis. As a result, ANT therapy must be withdrawn or suspended. The only pharmaceutical approved by the Food and Drug Administration (FDA) to prevent the development of anthracycline cardiomyopathy is dexrazoxane (DEX) [3,4,5]. The impact of its activity, however, is limited [6]. As a result, new solutions to minimize the risk of cardiomyopathy in ANT patients are continuously being investigated. Just a few substances with a possible protective effect established in tests conducted have been adopted in clinical trials to investigate the prevention of ANT cardiomyopathy. β-blockers and angiotensin-converting enzyme inhibitors (bisoprolol and ramipril) have been found to inhibit cardiac remodeling and improve mortality in patients with cardiac dysfunction and have been advocated for cardioprotection in cancer, which provides some confidence [7,8,9]. In this paper, we investigate the impact of carvedilol (CVD), a β-blocker, on the hearts of rats given DOX and DEX.

The pathogenesis of AIDC has been attributed mostly to the generation of reactive oxygen species (ROS) by the ANT-iron complex and the direct ANT redox cycle [1,6,10,11,12,13]. ROS generated in this manner, through molecular and gene damage, codes “a program” of mitochondrial failure, which has been clinically silenced throughout the years. Nevertheless, combining ANTs with numerous antioxidants failed to increase cardioprotection, raising doubts about the assumption that oxidative stress is the primary cause of congestive heart failure (CHF) [2,14,15]. Energy metabolism abnormalities, particularly mitochondrial ones, have been the focus of study since the beginning of research on AIDC. The body of data supporting this hypothesis is still growing to the present day. Doxorubicin has been proven to disrupt the energy metabolism of the heart muscle and cardiomyocytes [12,16,17], and more recently, the activation of molecular mechanisms leading to mitochondrial fragmentation has been discovered [18]. Alterations in the mitochondria have been identified at the ultrastructural, biochemical (decreased cytochrome c-oxidase (COX) activity), molecular (decreased concentration of COX II subunit mRNA), and genetic levels (decreased number of copies and mtDNA oxidation) [5,19].

The chelation of iron ions, and hence, the inhibition of ROS production was linked to the preventative effects of dexrazoxane reported in the clinic [15]. Nevertheless, accumulating data suggest that this mechanism is not important, as other iron chelators have little protective effect [20]. The major protective mechanism of DEX in AIDC has recently been demonstrated [4]. ANT interrupts TOP2β’s normal catalytic cycle, resulting in DNA double-strand breaks, which can lead to cardiomyocyte death. DEX was shown to bind topoisomerase 2β (TOP2β), preventing ANT binding [4,21,22,23].

Few compounds have entered clinical trials for the registration of drugs with anti-AIDC activity. CVD (NCT04023110) is one of them. It is a β-blocker with an antioxidant feature. CVD was shown in two separate randomized clinical studies to prevent AIDC [24,25]. Past research has demonstrated that CVD had a favorable effect on cardiac mitochondria in in vitro, ex vivo, and in vivo models [13]. Carvedilol, in particular, is considered to act as an inhibitor of mitochondrial complex I, which is recognized as a donor of NADH for the ANT redox cycle and, as a result, the cause of AIDC [24,25,26,27]. CVD was shown to be more effective than propranolol in preventing DOX-induced cardiomyopathy [26].

Although DEX and CVD have many diverse protective mechanisms, we may anticipate that their combined effect in AIDC prevention ought to be synergistic. Therefore, we made a hypothesis that combining DEX and CVD treatment in DOX-treated rats could be more protective than DEX alone. Clinical, functional (cardiac echocardiography (ECHO)), morphological, and biochemical research was performed. Owing to the DOX-induced energy metabolism abnormalities, an additional screening of the expression of chosen genes linked to the regulation of energy metabolism pathways in the myocardium was conducted.

## 2. Results

To perform the experiment, rats were divided into two main groups based on the time of euthanasia (groups I, and groups II). Groups I were euthanized one week after the end of the drug administration (control group, CTR_I_; experimental groups: DOX_I_, DOX + DEX_I_, DOX + CVD_I_, DOX + DEX + CVD_I_), and groups II were euthanized eleven weeks after the end of the drug administration (control group, CTR_II_; experimental groups: DOX_II_, DOX + DEX_II_, DOX + CVD_II_, DOX + DEX + CVD_II_). To illustrate the design of the experiment, we have prepared Figure 1. Details on the methodology and the exact division into groups can be found in Section 4.2—Experimental Design.

Regarding the purpose of the study, the differences between the DOX + DEX + CVD and DOX + DEX or DOX + CVD groups received the most attention when analyzing the results.

### 2.1. Echocardiography

Cardiomyopathy symptoms were assessed using cardiac ECHO (Figure 2). Only data from groups II were provided in order to properly evaluate changes in the ECHO of the heart in the same rats over a 21-week period. Detailed statistical data can be found in the Appendix A.

There were no significant changes in the ejection fraction in rats receiving DOX alone in the 11th week. Surprisingly, when CVD was delivered in addition to DOX, the value of this parameter was significantly reduced compared to CTR_II_. Consistent with the dose-matching assumptions, a drop in the ejection fraction was observed in the DOX_II_ group but not in the DOX + DEX_II_ group. Furthermore, contrary to the predictions, adding CVD to the DOX + DEX-treated rats resulted in a decrease in the ejection fraction. No significant changes were identified in the rats receiving DOX alone in the other parameters examined by ECHO, namely the left ventricular end-diastolic diameter, left atrial diameter, ascending aorta diameter, and fractional shortening. It was discovered that in the DOX + CVD-treated groups, the values of the left atrial diameter and the ascending aorta diameter parameters significantly decreased (Appendix A).

### 2.2. Histological Staining

Table 1 demonstrates the presence and intensity of the morphological changes in the rats’ hearts after the study treatment.

H + E staining revealed an increased distribution of eosinophils and the infiltration of mononuclear cells in the DOX, DOX + DEX + CVD, and DOX + CVD groups (Figure 3). Van Gieson staining revealed an increase in collagen deposition surrounding intracoronary blood vessels in the following groups of rats subjected to the study treatment: DOX_I_, DOX_II_, DOX + DEX + CVD_II_, and DOX + CVD_II_ (Figure 4). In both time periods, DEX avoided these alterations induced by DOX alone (DOX + DEX_I_; DOX + DEX_II_). This is consistent with the changes observed by ECHO. In either group, there were no symptoms of necrosis in the microscopic picture. Representative histopathology pictures are shown below (Figure 3 and Figure 4).

### 2.3. Biochemical Analyses

Biochemical analyses for the markers of cardiomyocyte necrosis (cardiac troponin I, cTnI) and ventricular myofiber diastole (brain natriuretic peptide, BNP) in the blood serum showed that they were more vulnerable to alterations. DOX significantly influenced the cTnI (DOX_I_, *p* < 0.0001; DOX_II_, *p* < 0.0001) and BNP (DOX_I_, *p* < 0.0487; DOX_II_, *p* < 0.0128) serum levels in the DOX-administered groups (Table 2). The addition of DEX clearly protected the heart muscle from damage. The results of the combinations performed were statistically insignificant in comparison to the control groups. The increase in the cTnI level of the DOX-treated rats was not prevented by CVD premedication (DOX + CVD_I_, *p* < 0.0005; DOX + CVD_II_, *p* < 0.0025). In the case of BNP, an elevated level was found in the 21st week of the experiment (DOX + CVD_II_, *p* < 0.0001). The administration of the combination premedication had no effect on the levels of cTnI or BNP in the 11th week of the study. In the 21st week, there was a significant increase in the BNP levels (DOX + DEX + CVD_II_, *p* < 0.0443).

### 2.4. The Assessment of Gene Expression

Selected genes encoding proteins involved in carbohydrate, lipid, and protein metabolism, which are important in myocardial energy metabolism, were evaluated. Gene expression analysis was carried out in the 11th and 21st weeks of the study. The major questions in these investigations were whether and how DOX modifies the expression of the examined genes, as well as whether and how adding CVD to the therapy regimen changes the gene expression in the hearts of rats treated with DOX and DEX at the same time. The goal of these findings was to gain a better understanding of the early molecular changes in ANT cardiomyopathy and to determine whether adding CVD to DEX will provide greater protection against cardiotoxicity in DOX-treated rats.

Changes in the gene expression were observed in the hearts of the DOX-treated rats in most functional panels, including genes regulating glucose transport, glucose metabolism, gluconeogenesis, glutaminolysis, pyruvate transport, lipid transport, and oxidation. The exact data are presented in Table 3.

In the animals euthanized in the 11th week of the study, the *Myc* gene, involved in glycolysis control, was significantly lower after the DOX treatment (*p* < 0.001). This decrease was accompanied by the down-expression of *Myc*-targeted genes (*Slc2a1* (*p* < 0.001); *G6pc1* (*p* < 0.001); *Gls* (*p* < 0.001)). The combination of DOX and DEX also significantly decreased *Myc* mRNA expression (*p* < 0.05) in comparison to the control group. It also influenced other glycolysis-related genes: *Slc2a1* (*p* < 0.001), *Slc2a4* (*p* < 0.001), *Ldha* (*p* < 0.001), and *G6pc1* (*p* < 0.001). Significant differences were observed after the DOX pretreatment with DEX in comparison to the DOX alone group: *Myc* (*p* < 0.001), *Slc2a4* (*p* < 0.001), and *Ldha* (*p* < 0.001). No statistically significant differences were observed for the *Slc2a1* (*p* = 0.187) and *G6pc1* (*p* = 0.066) genes. The combination of DOX and CVD significantly reduced *Slc2a1* (*p* < 0.001), *Slc2a4* (*p* < 0.001), *G6pc1* (*p* < 0.001), and *Ldha* (*p* < 0.001) expression in comparison to the CTR_I_ group. It had no impact on *Myc* (*p* < 0.275) or *Gls* (*p* < 0.973) mRNA expression. The DEX and CVD pretreatment decreased *Slc2a1* (*p* < 0.001), *G6pc1* (*p* < 0.001), and *Gls* (*p* < 0.001) expression and did not change *Myc* (*p* > 0.999), *Slc2a4* (*p* > 0.999), or *Ldha* (*p* = 0.7958) expression. The results of the study conducted on the rats euthanized in the 21st week showed no significant effect of the treatment on *Myc* or *Ldha* expression. The DOX treatment caused the down-expression of the *Slc2a1* (*p* < 0.001), *Slc2a4* (*p* < 0.001), *G6pc* (*p* < 0.001), and *Gls* (*p* < 0.001) genes. The DOX pretreatment ‘prevented’ such changes in almost all cases, but not for *Slc2a4* in the DOX + DEX + CVD_II_ (*p* < 0.001) or DOX + DEX_II_ (*p* < 0.001) groups, *G6pc1* in DOX + DEX_II_ (*p* < 0.001), or *Gls* in DOX + DEX + CVD_II_ (*p* < 0.001).

We further examined which genes contributed to fatty acid (FA) uptake and oxidation. The first was the *Prkaa2* gene. In the 11th week of the experiment, DOX down-regulated *Prkaa2* expression (*p* < 0.01). The DOX pretreatment significantly increased *Prkaa2* expression in other groups (*p* < 0.001). There was no statistical significance in *Prkaa2* expression after the following 10 weeks. Further, we examined the changes in the expression of *Cpt2.* DOX administration has no influence on *Cpt2* expression in rats euthanized in the 11th week. DEX pretreatment caused significant upregulation of this gene in the DOX + DEX_I_ group (*p* < 0.001, vs. CTR_I_). In the other groups, no changes were found. In the 21st week, DOX slightly reduced the *Cpt2* expression level. Another gene responsible for the transport of long-chain FAs, *Slc27a1*, was up-regulated after DOX (*p* < 0.001) as well as DEX + DOX (*p* < 0.001) administration in the 11th week of the study. DOX-induced effects persisted after the following 10 weeks (*p* < 0.001). DEX pretreatment lowered *Slc27a1* expression (DOX + DEX_II_, (*p* < 0.001)_;_ DOX + DEX + CVD_II_, (*p* < 0.001)). CVD pretreatment had no impact on *Slc27a1* expression. In this study, neither DOX nor its pretreatment showed an influence on *Cd36* expression. The *Ppargc1a* expression level was increased in the DOX_I_ (*p* < 0.05) and DOX + DEX_I_ (*p* < 0.001) groups and significantly decreased in the DOX_II_ group (*p* < 0.001). In our study, *Mpc1* expression was not affected by the DOX treatment in the DOX_I_ group but slightly increased in the DOX_II_ group (*p* < 0.001). The DOX pretreatment significantly increased the *Mpc1* mRNA level in the DOX + DEX_I_ and DOX + CVD_I_ groups (*p* < 0.001), but after the following 10 weeks, its expression level returned to the CTR_II_ level in these groups. DOX did not affect the expression of *Got2* in the DOX_I_ group. However, there was a slight change observed in the DOX_II_ group (*p* < 0.001). The DEX and CVD pretreatment caused significant *Got2* overexpression in the DOX + DEX_I_ (*p* < 0.001), DOX + CVD_I_ (*p* < 0.001), and DOX + DEX + CVD_I_ groups (*p* < 0.05). Finally, the DOX treatment significantly increased *Acadm* expression (DOX_I_, *p* < 0.001; DOX_II_, *p* < 0.001). The simultaneous administration of DEX and DOX led to a 40-fold expression increase (*p* < 0.001), CVD and DOX led to a 30-fold expression increase (*p* < 0.001), and the DEX, CVD, and DOX treatment led to a 10-fold expression increase (*p* < 0.001). After the following 10 weeks, *Acadm* expression returned to the CTR_II_ levels.

### 2.5. Macroscopic Clinical Observations

Macroscopic clinical assessment revealed systemic toxicity in the rats given DOX and DOX in combination with CVD (Table 4). Ascites were found in the DOX_I_ and DOX + CVD_I_ groups. Blood purulent ascites were detected in DOX_II_ and DOX + CVD_II_. Furthermore, rats in both the DOX_I_ and the DOX + CVD_I_ groups had enlarged livers. Previous fatalities were observed in the DOX_II_ (*n* = 5/10), and DOX + CVD_II_ (*n* = 4/10) groups. There were no signs of clinically significant toxicity in the DOX + DEX or DOX + DEX + CVD groups.

## 3. Discussion

Here, we performed functional (cardiac ECHO), morphological, biochemical, and molecular research on rats to investigate whether combining DEX and CVD pretreatments was favorable and prevented ANT-induced cardiomyopathy. Apart from the ejection fraction parameter level in the 21st week, no statistically significant differences were found in the ECHO parameters, BNP, or cTnI levels between the DOX + DEX + CVD and DOX + DEX groups. The histopathological examinations revealed that the DOX + DEX + CVD group had a higher frequency of pathological features than the DOX + DEX group. These changes are not reflected in the gene expression; the addition of CVD (DOX + DEX + CVD group) normalized the altered expression of the vast majority of the marked genes in the DOX + DEX group (vs. CTR).

Since the 1980s, the free radical theory has appeared to be the most credible explanation for AIDC pathomechanism [1]. In laboratory experiments during the following few decades, highly encouraging results of antioxidant chemicals’ protective action were obtained. Unfortunately, clinical studies have not confirmed the effectiveness of antioxidants in AIDC [2,14,15,28,29,30]. Despite this, the FDA has approved only one iron chelator, DEX, for AIDC protection. Originally, the principal protective mechanism of DEX was related to restricting the pool of iron interacting with ANT, resulting in a compound several hundred times more effective than ANT alone in producing free radicals and oxidative stress [1,10]. However, recent evidence suggests that the main protective mechanism of DEX in AIDC is to prevent ANT from inhibiting topoisomerase 2β [4,21,22,23]. Nonetheless, the importance of oxidative stress in AIDC cannot be overstated. Clinical trials on the licensing of medications indicating a preventative benefit in AIDC included, among other things, CVD, a β-blocker with an antioxidant component [9]. CVD, on the other hand, has been proven to prevent AIDC in two separate randomized clinical studies [24,25]. Therefore, the aim of this study was to investigate the predicted synergy of the protective effect in AIDC (DEX combined with CVD—substances with a varied mechanism that decreases oxidative stress, as well as a mechanism that prevents TOP2β inhibition by DOX).

### 3.1. DOX Effects

In rats, the cumulative DOX-induced AIDC model is well recognized. However, in addition to the primary goal of the study, we intended to further optimize the AIDC model. Cardiomyopathy should occur following the termination of treatment, just as it happens in a clinical situation. The systemic toxic impact emerged after 7 weeks in our prior experiment with a dosage of 2.0 mg/kg b.w. once a week for 10 weeks [31]. Significant mortality was seen in a comparable model at a dose of 2.5 mg for 10 weeks [32]. Based on our own experiments and the literature, we determined that the dosing method used in the experiment (1.6 mg/kg b.w. once a week for 10 weeks) would be optimal, causing cardiomyopathy not only during the period of DOX administration (until the 10th week of the study) but also over the following 10 weeks, with minimal overall toxicity.

Changes in gene expression were detected in almost all specified pathways linked to the energy balance in rats given DOX alone (Table 3). There were no alterations detected in selected genes involved in FA metabolism (*Cd36*) and pyruvate-to-lactate conversion (*Ldha*). It is difficult to tell which changes are favorable (adaptive) and which are unfavorable, damaging, or crucial for toxicity. However, it appears that the negative effect is more likely to be attributed to decreases in gene expression (*Slc2a1*, *Slc2a4*, *G6pc*, *Gls*, *Cpt2*, *Ppargc1*, and *Myc*) than to increases (*Mpc1*, *Got2*, *Slc27a1*, and *Acadm*). Increased gene expression is just the first step toward protein synthesis, which means that overexpression does not imply an increase in protein synthesis. A drop in gene activity, on the other hand, is associated with a considerably higher likelihood of a negative outcome. Reduced protein synthesis results in a weakening or loss of the function-associated protein, i.e., a harmful effect on the cell. DOX was observed to significantly increase the levels of cTnI in the blood serum, which might imply cardiomyocyte necrosis. Despite the fact that the general histological image in this group of rats suggests the establishment of pathogenic alterations, including mononuclear cell infiltration, an increased distribution of eosinophils, and collagen deposition, no necrotic changes were seen. This demonstrates that necrotic alterations in cardiomyocytes were still subtle, as they were seen at the biochemical level but not at the tissue level. The macroscopic appearance of the heart, particularly in the 21st week of the experiment, revealed remodeling and the likelihood of decreasing the heart’s contractility (thinning of the walls). This is supported by a rise in the serum BNP levels (Table 2) as well as a decrease in the ejection fraction parameters in the 21st week of the research (Figure 2). To summarize, the obtained results confirmed that the applied research model of ANT cardiomyopathy was positively validated, and the results also indicate unfavorable changes at the level of genes involved in the energy balance of the cell, responsible for intracellular glucose transport, gluconeogenesis, glutaminolysis, and lipid oxidation. According to the assumptions, it was also conceivable to choose the DOX dose such that cardiomyopathy did not develop during administration up to the 11th week, but only after it was finished, from the 11th to the 21st week. This is demonstrated by the lack of changes in the ECHO parameters in the 11th week of the research and a substantial drop in the ejection fraction at the end of the 21st week.

### 3.2. Changes in the DOX + DEX + CVD vs. DOX + DEX Groups

As previously stated, in the assessment of cardiac function utilizing ECHO, significant changes in the groups receiving DOX alone were detected solely for the ejection fraction value in the 21st week (Figure 2). There were no statistically significant variations in the left ventricular end-diastolic diameter, left atrial diameter, ascending aorta diameter, or fractional shortening in the ECHO examination between the DOX + DEX + CVD and DOX + DEX groups, indicating that CVD had no influence on DOX + DEX in terms of these parameters. There were substantial changes in the ejection fraction between the DOX + DEX + CVD and DOX + DEX groups; however, giving CVD to DOX + DEX rats increased the drop in the ejection percentage; therefore, this had a negative impact. As a result, the assumption that a synergy of the preventive effects of DEX and CVD in AIDC might be predicted was not validated.

There was no significant difference in the cTnI between the DOX + DEX + CVD and DOX + DEX groups in comparison to the controls (Table 2). On the other hand, DOX-induced change normalization of the cTnI was detected in both groups. It is quite likely that the DEX, not CVD, was responsible for normalizing these alterations. This concept is supported by the lack of changes in the cTnI levels between the DOX + CVD and DOX groups, indicating that CVD had no protective effect. In contrast, despite the fact that CVD had no effect on the frequency of the early beginning of the left ventricle ejection fraction decline, there was an observed substantial reduction in the cTnI levels and diastolic dysfunction in patients receiving DOX [26]. Normalization of the DOX-induced changes was observed in all pretreated groups for BNP in the 11th week of the study (Table 2). A comparison of the DEX and CVD groups, however, reveals that DEX had a greater protective impact. The supposed synergism of the preventive effects of DOX and CVD is difficult to verify because DEX alone normalized the elevation in BNP produced by DOX. A similar result was found for BNP in the 21st week when DEX alone was found to be more protective against DOX-induced abnormalities than CVD alone or CVD in conjunction with DEX. A histopathological examination revealed that the DOX + DEX + CVD group had a higher frequency and intensity of pathological features than the DOX + DEX group (Table 1). This demonstrates not only a lack of protection but also an unfavorable effect of CVD at the tissue level in DOX + DEX-treated rats.

A contrary effect of CVD was established for the examined genes involved in myocardial cell energy balance. The inclusion of CVD (DOX + DEX + CVD group) normalized the aberrant expression of the vast majority of indicated genes in the DOX + DEX group (vs. control, Table 3). The effect of such normalization on the expression of *Myc*, *Slc2a1*, *Got2*, *G6pc*, *Mpc1*, *Ldha*, *Cpt2*, *Prkaa2*, *Acadm*, and *Gls* was demonstrated in the 11th week of the research. However, by the 21st week of the study, only the expression of the *G6pc* and *Gls* genes had normalized. This suggests that CVD’s protective effect against the alterations in the gene expression reported in the DOX + DEX group was just transitory and faded over time once the treatment ended.

In terms of macroscopic observations, general toxicity, ascites, and an enlarged liver were discovered after euthanasia in almost all rats in the DOX_I_ and DOX + CVD_I_ groups (Table 4). Blood purulent ascites and prior fatalities were reported in the DOX_II_ and DOX + CVD_II_ groups, which may suggest organ damage. This also shows that CVD protection is lacking in DOX-induced systemic toxicity effects. However, no ascites or previous deaths were observed in the rats pretreated with DEX (DOX + DEX and DOX + DEX + CVD groups), demonstrating the protective effect of DEX alone against these disorders and the limited role of CVD. This finding cannot be attributed to DEX’s function in the prevention of DOX-induced ejection fraction decline (congestive heart failure) because the DOX + DEX + CVD_II_ group had the lowest ejection fraction parameter (mean 64.60%; mean CTR_II_, 80.30%) despite the absence of ascites or prior fatalities. As a result, the mechanism of DEX protection in the prevention of ascites and early deaths must be distinct, and it is most likely connected to poor protein synthesis in the liver [33]. It is worth noting that we previously detected ascites in a comparable animal but with a larger dosage of DOX (unpublished findings). Ascites, on the other hand, have not been recorded as a side effect of DOX in people.

Overall, the data show that DEX played a significant role in protecting against functional and biochemical alterations (cTnI and BNP) produced by DOX. At the functional (ECHO) and biochemical levels, there was no advantageous synergism between DEX and CVD in the cardioprotection of alterations generated by DOX, and at the tissue level, CVD employed for protection had an adverse impact when combined with DEX. The normalization of the expression of most CVD-marked genes was temporary. It is difficult to make non-speculative inferences from these very first outcomes of molecular research using multiple vectors of change. This feature may be clarified if future research includes a larger panel of genes from particular energy pathways and the results are connected with functional, biochemical, and morphological alterations. We are aware of the limitations of the conducted molecular research, which was the selection of single genes representing specific biochemical pathways. A microarray analysis that takes into account a wide range of genes would certainly give a broader picture. However, in this study, the functional, biochemical, and histological results show that there is no reason to recommend the simultaneous administration of DEX and CVD in DOX cardiotoxicity. It is not justified to extend research at the molecular level due to the lack of synergy of the protective effect of DEX and CVD in DOX-induced cardiomyopathy at the clinical level.

## 4. Materials and Methods

### 4.1. Animals

A total of a hundred male 8-week-old Wistar rats were obtained from the Experimental Medicine Center, Medical University of Lublin (Lublin, Poland). All animals were maintained under controlled environmental conditions (temperature range: 22 ± 3 °C; relative humidity: 50 ± 5%; a 12 h light/dark cycle) throughout the experimental period. The rats had unlimited access to drinking water and a conventional rodent diet. All procedures were carried out between 9 a.m. and 3 p.m. The animal study protocol was approved by the Local Ethical Committee for Animal Experiments based at the University of Life Sciences in Lublin, Poland (protocol code 123/2018 and date of approval: 3 December 2018). All experimental animal procedures were conducted under the European Committee Directive for Care and Use of Laboratory Animals (2010/63/EU). The animals were under the constant supervision of the veterinarian. Every attempt was made to reduce harm.

### 4.2. Experimental Design

The rats were kept for 7 days before the beginning of the experiment for acclimatization purposes. Then, the animals were assigned randomly into ten study groups depending on the type of administration and the time of their euthanasia. Each experimental group consisted of ten rats at the beginning of the study. Some of the animals (groups I) were euthanized in the 11th week, one week after the end of the administration period (1st control group, CTR_I_; experimental groups: DOX without pretreatment, DOX_I_; DEX and CVD pretreatment 30 min prior to DOX, DOX + DEX + CVD_I_; DEX pretreatment 30 min prior to DOX, DOX + DEX_I_; CVD pretreatment 30 min prior to DOX, DOX + CVD_I_) and the rats’ samples were used for morphological, biochemical, and genetic investigations. For the other animals (groups II), cardiac ECHO was performed at the 1st, 11th, and 21st weeks of study, and they were euthanized in the 21st week, 10 weeks after the end of the administration (2nd control group, CTR_II_; experimental groups: DOX without pretreatment, DOX_II_; DEX and CVD pretreatment 30 min prior to DOX, DOX + DEX + CVD_II_; DEX pretreatment 30 min prior to DOX, DOX + DEX_II_; CVD pretreatment 30 min prior to DOX, DOX + CVD_II_). The rats’ samples were used for the morphological, biochemical, and genetic investigations. During the experiment, the number of animals in the two groups changed (DOX_II_, *n* = 5; DOX + CVD_II_, *n* = 6). A detailed description of the groups is provided in Table 5. DOX, DEX, and CVD were purchased from Merck (Darmstadt, Germany). All solutions were prepared in a volume of 0.01 mL/g body weight immediately before administration. The rats were euthanized under 3.5% isoflurane anesthesia by decapitation, and the rats’ hearts were then harvested for pathology and molecular studies. The blood was collected and centrifuged to obtain the serum for further biochemical analysis.

### 4.3. Echocardiography

Two-dimensional (2D) echocardiograms (ECHOs) were carried out during the investigation using the ESAOTE MyLab 25 GOLD ultrasound machine (Esaote, Genoa, Italy). The ejection fraction, left ventricular end-diastolic diameter, left atrial diameter, ascending aorta diameter, and fractional shortening were measured before starting the DOX treatment, during treatment, and at the end of treatment. Only data from groups II were provided in order to properly evaluate changes in the ECHO of the heart in the same rats over a 21-week period.

### 4.4. Histological Staining

Pieces of the apex of the heart were collected from the left ventricle of each individual in buffered 10% formalin (pH = 7.4), and processed to paraffin blocks. Four-micrometer slides were cut on the microtome and stained with hematoxylin and eosin, and picric acid and acid fuchsin (van Gieson), to visualize changes in the myocardium caused by the study treatment. The slides were evaluated under a light microscope by an experienced pathologist. The sections were evaluated for the presence and severity of inflammation and fibrosis. Histological changes were classified as follows: −, no observed changes; + changes of minor intensity; ++ moderate changes. Figure 3 and Figure 4 were made using a Nikon Eclipse TiE phase contrast microscope (Nikon, Tokyo, Japan)and Nikon’s NIS-Elements BR 3.2.Ink imaging software with auto-white balance and auto-contrast, with merging carried out in Inkscape’s vector graphics editor (Inkscape Project. (2020). Inkscape. Retrieved from https://inkscape.org (accessed on 17 April 2023)).

### 4.5. Biochemical Analysis

Cardiac Troponin I (cTnI) was analyzed in the rats’ blood serum using a Sandwich enzyme immunoassay ELISA Kit for cTnI (Cloud-Clone Corp., Katy, TX, USA) according to the manufacturer’s instructions. The brain natriuretic peptide (Bnp) concentrations were analyzed in the rats’ blood serum using RayBio Mouse/Rat Bnp EIA Kit (RayBiotech Life, Inc., Peachtree Corners, GA, USA) according to the manufacturer’s instructions. Color changes were measured spectrophotometrically using the PowerWave microplate spectrophotometer (BioTek Instruments, Winooski, VT, USA) at a wavelength λ = 450 nm, and the Bnp concentrations were determined in relation to standard curves.

### 4.6. Molecular Studies (RT-PCR Analysis)

Rat heart mRNA was used to perform an expression study on the genes involved in cell metabolism. Following the manufacturer’s instructions, the total RNA was extracted from 50 mg of tissue sections taken from the left ventricle using TRIzol reagent (Invitrogen, Carlsbad, CA, USA). A MaestroNano NanoDrop spectrophotometer was used to determine the concentration and purity of the isolated RNA (Maestrogen, Hsinchu, Taiwan). For further studies, high-purity RNA was used (A260/280: 1.8–2.0). Then, using a cDNA reverse transcription kit, cDNA synthesis was performed (Applied Biosystems, Foster City, CA, USA). Reaction conditions: 25 °C for 10 min, 37 °C for 120 min, and then 85 °C for 5 min. Real-time PCR was used to quantify the relative expression of the investigated genes using the high-throughput SmartChip MyDesign Chip system (WaferGen Bio-Systems, Fremont, CA, USA) using PowerUp SYBR Green Master Mix (Applied Biosystems, Foster City, CA, USA). Four technical replications of each reaction were run (*n* = 28 trials from 7 animals per group). The reaction profile: 95 °C for 2 min, 45 cycles: 95 °C for 15 s, 57 °C for 15 s, and 72 °C for 1 min; melt curve 0.4 °C/s up to 97 °C. Before calculating the ΔΔCt and determining the fold change in the mRNA levels, a data quality analysis of the samples was conducted based on the amplification, Tm, and Ct values to screen out any outliers. The data were normalized using the *Rpl32* and *Tbp* housekeeping genes and presented as the mean relative quantification (RQ).
RQ = (2^−ΔΔCt^)

The capacity of the heart to generate energy from free FAs is well established. However, if necessary, it may oxidize a variety of other fuels, such as glucose, lactate, ketones, and amino acids [34,35]. Thus, abnormalities of aerobic respiration generated by DOX may result in the activation of alternate, anaerobic energy generation routes, namely glycolysis. Disruptions in the FAs and glucose transport, on the other hand, may result in intracellular glucose production via gluconeogenesis and early activation of amino acid synthesis. The mRNA studies were used for the initial recognition of changes in the gene expression of a given signaling pathway responsible for the transformation of energy metabolism. As a result, we focused our research on particular genes that indicate pathways whose regulation may be altered as a result of DOX-induced mitochondrial diseases. We were guided by the importance of enzymes in a specific pathway when determining the genes, favoring those with irreversible reactions. Detailed information on the genes and primer sequences used in this study is provided in Table 6.

### 4.7. Macroscopic Clinical Observations

After the animals were euthanized, macroscopic findings of general toxicity were determined. Mortality, ascites presence, and liver appearance were evaluated using the following objective scoring: −, no changes; +, changes of minor intensity; ++, moderate changes; +++, changes of major intensity.

### 4.8. Statistical Analysis

Statistical analysis was performed using GraphPad Prism version 6.04 for Windows, GraphPad Software, www.graphpad.com (accessed on 13 February 2023). Tukey’s post-hoc tests (HSD and Spjotvolla–Stoline) were applied in combination with a one-way ANOVA for the statistical analysis. The data were calculated as the mean ± SD. When the *p*-value was under 0.05, differences among the groups were regarded as statistically significant.

## Figures and Tables

**Figure 1 ijms-24-10202-f001:**
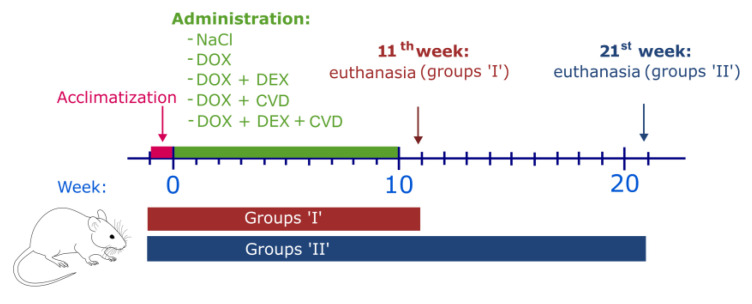
The study scheme. CVD, carvedilol; DOX, doxorubicin; DEX, dexrazoxane.

**Figure 2 ijms-24-10202-f002:**
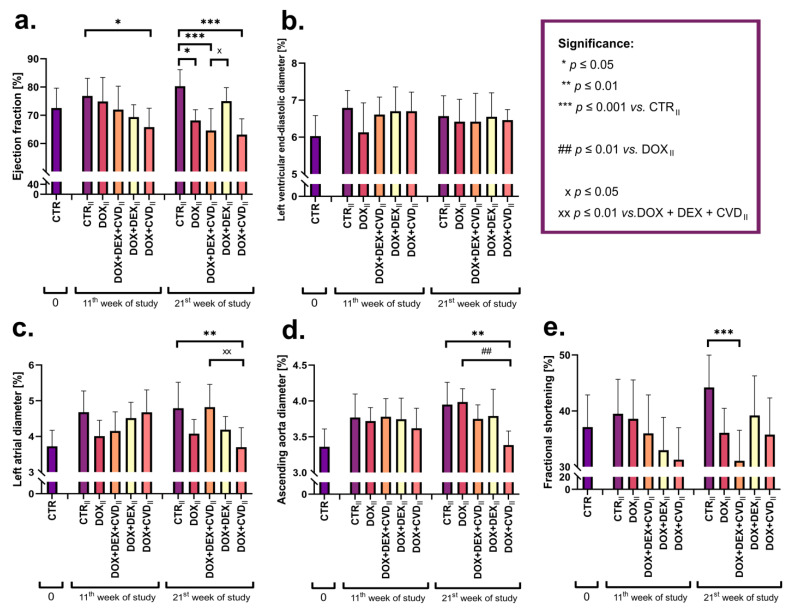
Echocardiography (ECHO) results of the following parameters: (**a**) ejection fraction; (**b**) left ventricular end-diastolic diameter; (**c**) left atrial diameter; (**d**) ascending aorta diameter; (**e**) fractional shortening in all study groups. Data are expressed as means ± standard deviation. 0, ECHO results after acclimatization and before the beginning of the treatment. CTR, groups II before the treatment; CTR_II_, control II; CVD_II_, carvedilol II; DOX_II_, doxorubicin II; DEX_II_, dexrazoxane II). Significance: * *p* ≤ 0.05; ** *p* ≤ 0.01, *** *p* ≤ 0.001 vs. the control group; ## *p* ≤ 0.01 vs. the DOX group; x *p* ≤ 0.05; xx *p* ≤ 0.01 vs. the DOX + DEX + CVD group.

**Figure 3 ijms-24-10202-f003:**
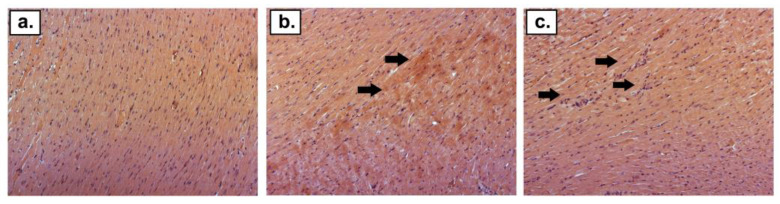
H + E staining. (**a**) Normal histopathology picture of the rat heart in the CTR_II_ group; (**b**) increased distribution of eosinophils (black arrows) in the DOX_II_ group; (**c**) mononuclear cell infiltration (black arrows) in the DOX + CVD_II_ group magnification ×150.

**Figure 4 ijms-24-10202-f004:**
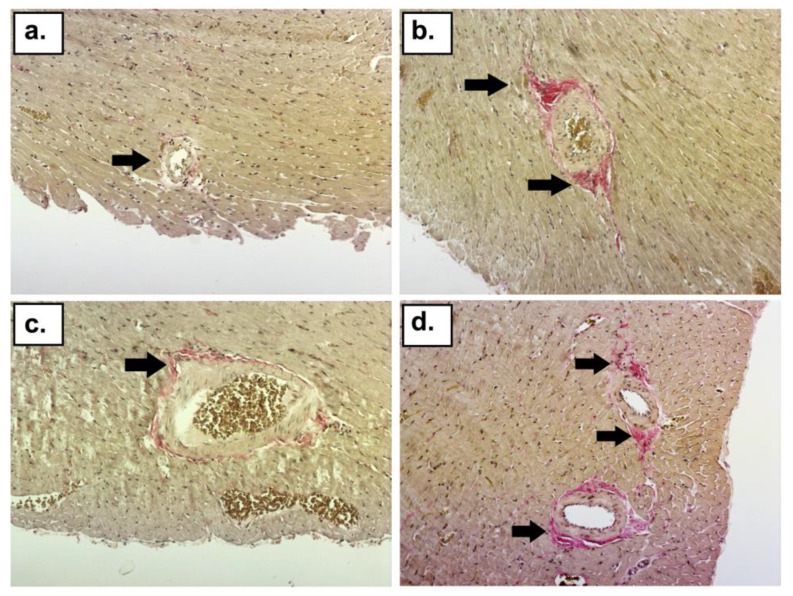
Collagen deposition surrounding intracoronary arterioles after study treatment. Sections of heart tissue were stained with van Gieson stain, demonstrating collagen (red), cardiac muscle (yellow), and nuclei (black). (**a**) Normal degree of collagen surrounding an arteriole (black arrow) in the CTR_I_ group; (**b**) collagen more intensely stained around the arteriole extending into the surrounding myocardium (black arrows) in the DOX_II_ group; (**c**) normal degree of collagen surrounding an arteriole (black arrow) in the DOX + DEX_II_; (**d**) collagen less intensely stained around the arteriole (black arrows) in the DOX + DEX + CVD_I_; magnification ×150.

**Table 1 ijms-24-10202-t001:** The presence and intensity of morphological changes in rats’ hearts after the study treatment.

Morphological Feature	CTR_I_	CTR_II_	DOX_I_	DOX_II_	DOX + DEX + CVD_I_	DOX + DEX + CVD_II_	DOX + DEX_I_	DOX + DEX_II_	DOX + CVD_I_	DOX + CVD_II_
Mononuclear cell infiltration	−	−	+	++	+	++	−	−	+	++
Distribution of eosinophils	−	−	+	+	+	+	−	−	+	+
Collagen deposition	−	−	+	+	−	+	−	−	−	+

−, No changes; +, changes of minor intensity; ++, moderate changes; CTR, control; CVD, carvedilol; DOX, doxorubicin; DEX, dexrazoxane.

**Table 2 ijms-24-10202-t002:** Results of serum cardiac troponin I (cTnI) and brain natriuretic peptide (BNP) levels in rats serum receiving doxorubicin (DOX), doxorubicin with dexrazoxane pretreatment (DOX + DEX), doxorubicin with carvedilol pretreatment (DOX + CVD), doxorubicin with dexrazoxane and carvedilol pretreatment (DOX + DEX + CVD) or NaCl (CTR) *i.p.*, euthanized in the 11th (I) or 21st (II) week of study. Data are presented as mean ± standard deviation (*n* = 10). Significance: * *p* ≤ 0.05; ** *p* ≤ 0.01, *** *p* ≤ 0.001 vs. the control group; # *p* ≤ 0.05; ### *p* ≤ 0.001 vs. the DOX group; ns, not statistically significant.

Parameter	Group	Mean Concentration ± SD (pg/mL)	*p*, vs. CTR	*p*, vs. DOX
cTnI	CTR_I_	4.388 ± 1.034	-	-
	DOX_I_	9.930 ± 3.517	<0.0001 ***	-
	DOX + DEX_I_	4.980 ± 1.223	0.9932 ns	<0.0001 ###
	DOX + CVD_I_	8.356 ± 2.289	0.0005 ***	0.5426 ns
	DOX + DEX + CVD_I_	5.310 ± 1.298	0.9361 ns	<0.0001 ###
	CTR_II_	4.530 ± 1.196	-	-
	DOX_II_	9.020 ± 1.749	<0.0001 ***	-
	DOX + DEX_II_	5.020 ± 1.440	0.9998 ns	<0.0001 ###
	DOX + CVD_II_	8.133 ± 0.954	0.0025 **	0.9618 ns
	DOX + DEX + CVD_II_	5.420 ± 1.767	0.9812; ns	0.0009 ###
BNP	CTR_I_	144.229 ± 25.695	-	-
	DOX_I_	279.011 ± 124.591	0.0487 *	-
	DOX + DEX_I_	154.690 ± 43.376	>0.9999 ns	0.0272 #
	DOX + CVD_I_	184.322 ± 53.796	0.9977 ns	0.2209 ns
	DOX + DEX + CVD_I_	151.240 ± 44.587	<0.9999 ns	0.0142 #
	CTR_II_	165.133 ± 50.883	-	-
	DOX_II_	299.300 ± 63.607	0.0128 *	-
	DOX + DEX_II_	224.413 ± 133.121	0.7707 ns	0.6242 ns
	DOX + CVD_II_	362.025 ± 283.230	<0.0001 ***	0.8196 ns
	DOX + DEX + CVD_II_	283.230 ± 120.987	0.0443 *	>0.9999 ns

**Table 3 ijms-24-10202-t003:** Results of serum gene expression levels in rats’ hearts receiving doxorubicin (DOX), doxorubicin with dexrazoxane pretreatment (DOX + DEX), doxorubicin with carvedilol pretreatment (DOX + CVD), doxorubicin with dexrazoxane and carvedilol pretreatment (DOX + DEX + CVD), or NaCl (CTR) euthanized in the 11th (I) or 21st (II) week of study. Data are presented as mean relative quantification (RQ) ± standard deviation (*n* = 8). Significance: * *p* ≤ 0.05; ** *p* ≤ 0.01, *** *p* ≤ 0.001 vs. the control group; # *p* ≤ 0.05; ## *p* ≤ 0.01, ### *p* ≤ 0.001 vs. the DOX group; x *p* ≤ 0.05; xx *p* ≤ 0.01, xxx *p* ≤ 0.001 vs. the DOX + DEX + CVD group (one-way ANOVA with Tukey’s post-hoc test).

Group		CTR	DOX	DOX + DEX + CVD	DOX + DEX	DOX + CVD
Gene	Time	Mean RQ ± SD				
*Myc*	I	1.010 ± 0.145	0.620 ± 0.149 ***	1.004 ± 0.172 ###	0.878 ± 0.139 * ###	0.918 ± 0.189 ###
	II	1.002 ± 0.077	1.012 ± 0.077	0.938 ± 0.100 #	0.987 ± 0.112	0.992 ± 0.077
*Slc2a1*	I	1.006 ± 0.112	0.542 ± 0.106 ***	0.659 ± 0.164 *** #	0.462 ± 0.127 *** xxx	0.612 ± 0.118 ***
	II	1.005 ± 0.155	0.577 ± 0.106 ***	0.942 ± 0.147 ###	0.972 ± 0.191 ###	0.982 ± 0.191 ###
*Slc2a4*	I	1.010 ± 0.147	1.016 ± 0.106	1.018 ± 0.122	1.937 ± 0.545 *** ### xxx	1.545 ± 0.422 *** ### xxx
	II	1.006 ± 0.113	0.589 ± 0.203 ***	0.689 ± 0.203 ***	0.783 ± 0.203 *** ##	1.056 ± 0.113 ### xxx
*G6pc1*	I	1.016 ± 0.185	0.396 ± 0.149 ***	0.724 ± 0.153 *** ###	0.517 ± 0.171 *** xxx	0.506 ± 0.126 *** xxx
	II	1.000 ± 0.180	0.356 ± 0.149 ***	1.071 ± 0.221 ###	0.657 ± 0.300 *** ### xxx	0.900 ± 0.180 ### x
*Ldha*	I	1.012 ± 0.177	1.133 ± 0.156	1.085 ± 0.295	1.415 ± 0.265 *** ### xxx	1.457 ± 0.207 *** ### xxx
	II	1.076 ± 0.123	0.976 ± 0.123	0.989 ± 0.106	1.095 ± 0.159 #	1.149 ± 0.157 ### xxx
*Gls*	I	1.007 ± 0.130	0.560 ± 0.148 ***	0.787 ± 0.195 *** ###	0.978 ± 0.153 ### xxx	1.034 ± 0.132 ### xxx
	II	1.001 ± 0.190	0.412 ± 0.148 ***	1.680 ± 0.350 *** ###	1.233 ± 0.322 * ### xxx	1.034 ± 0.218 ### xxx
*Mpc1*	I	1.012 ± 0.155	1.262 ± 0.183	1.353 ± 0.242	3.003 ± 1.232 *** ### xxx	2.376 ± 0.753 *** ### xxx
	II	1.003 ± 0.144	1.402 ± 0.183 ***	1.001 ± 0.113 ###	1.019 ± 0.133 ###	1.023 ± 0.144 ###
*Cpt2*	I	1.021 ± 0.207	0.874 ± 0.175	1.101 ± 0.134 ###	1.412 ± 0.202 *** ### xxx	0.953 ± 0.211
	II	1.000 ± 0.211	0.802 ± 0.207 *	0.947 ± 0.329	0.963 ± 0.248	1.002 ± 0.207 #
*Got2*	I	1.016 ± 0.184	1.344 ± 0.206	1.819 ± 0.587 *	3.791 ± 1.821 *** ### xxx	2.911 ± 0.981 *** ### xx
	II	1.001 ± 0.161	1.444 ± 0.206 ***	0.991 ± 0.163 ###	0.993 ± 0.119 ###	1.013 ± 0.119 ###
*Cd36*	I	1.011 ± 0.093	0.967 ± 0.124	0.947 ± 0.119	1.056 ± 0.135 x	0.963 ± 0.112
	II	1.005 ± 0.119	1.105 ± 0.119	0.974 ± 0.129 #	1.101 ± 0.155 x	1.201 ± 0.155 *** xxx
*Slc27a1*	I	1.051 ± 0.347	1.795 ± 0.580 ***	1.223 ± 0.167 ###	1.563 ± 0.554 *** x	0.786 ± 0.214 ### xx
	II	1.008 ± 0.088	1.559 ± 0.081 ***	0.859 ± 0.081 *** ###	0.845 ± 0.189 *** ###	1.108 ± 0.088 * ### xxx
*Ppargc1a*	I	1.012 ± 0.161	1.277 ± 0.213 *	1.120 ± 0.199	1.456 ± 0.246 *** xx	1.022 ± 0.534 #
	II	1.000 ± 0.250	0.404 ± 0.161 ***	0.869 ± 0.161 ###	0.893 ± 0.236 ###	0.843 ± 0.236 ###
*Prkaa2*	I	1.011 ± 0.156	0.704 ± 0.177 **	1.401 ± 0.325 *** ###	1.870 ± 0.428 *** ### xxx	1.677 ± 0.365 *** ### x
	II	1.003 ± 0.095	1.053 ± 0.095	0.921 ± 0.091 ###	1.031 ± 0.123 xx	1.081 ± 0.123 xxx
*Acadm*	I	1.035 ± 0.184	4.833 ± 1.919 ***	10.016 ± 3.854 *** ###	42.179 ± 26.038 *** ### xxx	29.549 ± 12.296 *** ### xxx
	II	1.004 ± 0.151	4.493 ± 1.919 ***	0.673 ± 0.100 *** ###	1.038 ± 0.123 ###	1.104 ± 0.151 ###

**Table 4 ijms-24-10202-t004:** The macroscopic observations and clinical symptoms of rats euthanized in the 11th (I) or 21st (II) week of study.

Group	Clinical Symptoms		
	Presence of Ascites	Enlarged Liver	Mortality
CTR_I_	−	−	−
CTR_II_	−	−	−
DOX_I_	++	+	−
DOX_II_	+++	−	++
DOX + DEX + CVD_I_	−	−	−
DOX + DEX + CVD_II_	−	−	−
DOX + DEX_I_	−	−	−
DOX + DEX_II_	−	−	−
DOX + CVD_I_	++	+	−
DOX + CVD_II_	+++	−	++

−, No changes; +, changes of minor intensity; ++, moderate changes; +++, changes of major intensity; CTR, control; CVD, carvedilol; DOX, doxorubicin; DEX, dexrazoxane.

**Table 5 ijms-24-10202-t005:** The experimental administration design.

Symbol of Group	Type of Group	Administration	Euthanasia
CTR_I_	Control (*n* = 10)	0.01 mL 0.9% NaCl per g body weight *i.p.* administration once a week for 10 weeks	11th week
DOX_I_	Experimental (*n* = 10)	1.6 mg DOX per kg of body weight *i.p.* administration once a week for 10 weeks	11th week
DOX + DEX + CVD_I_	Experimental (*n* = 10)	1 mg CVD per kg of body weight *i.p.* administration 30 min prior DOX; 25 mg DEX per kg of body weight *i.p.* administration 30 min prior DOX; 1.6 mg DOX per kg of body weight *i.p.* administration once a week for 10 weeks	11th week
DOX + DEX_I_	Experimental (*n* = 10)	1.6 mg DOX per kg of body weight *i.p.* administration once a week for 10 weeks; 25 mg DEX per kg of body weight *i.p.* administration 30 min prior DOX	11th week
DOX + CVD_I_	Experimental (*n* = 10)	1.6 mg DOX per kg of body weight *i.p.* administration once a week for 10 weeks; 1 mg CVD per kg of body weight *i.p.* administration 30 min prior DOX	11th week
CTR_II_	Control (*n* = 10)	0.01 mL 0.9% NaCl per g body weight i.p. administration once a week for 10 weeks	21st week
DOX_II_	Experimental (*n* = 10)	1.6 mg DOX per kg of body weight *i.p.* administration once a week for 10 weeks	21st week
DOX + DEX + CVD_II_	Experimental (*n* = 10)	1 mg CVD per kg of body weight *i.p.* administration 30 min prior DOX; 25 mg DEX per kg of body weight *i.p.* administration 30 min prior DOX; 1.6 mg DOX per kg of body weight *i.p.* administration once a week for 10 weeks	21st week
DOX + DEX_II_	Experimental (*n* = 10)	1.6 mg DOX per kg of body weight *i.p.* administration once a week for 10 weeks; 25 mg DEX per kg of body weight *i.p.* administration 30 min prior DOX	21st week
DOX + CVD_II_	Experimental (*n* = 10)	1.6 mg DOX per kg of body weight *i.p.* administration once a week for 10 weeks; 1 mg CVD per kg of body weight *i.p.* administration 30 min prior DOX	21st week

CTR, control; CVD, carvedilol; DEX, dexrazoxane; DOX, doxorubicin.

**Table 6 ijms-24-10202-t006:** The symbols and names of the genes, GenBank reference sequence accession numbers, assay IDs, and the lengths of amplicon (bp).

Gene Name	Gene Symbol	Primer Sequence (5′ → 3′)		Product Size (bp)	NCBI Reference Sequence
		Left	Right		
Solute Carrier Family 2 Member 1	*Slc2a1*	GCC TGA GAC CAG TTG AAA GC	GAG TGT CCG TGT CTT CAG CA	154	NM_138827.1
Solute Carrier Family 2 Member 4	*Slc2a4*	GCTTCTGTTGCCCTTCTGTC	TGGACGCTCTCTTTCCAACT	166	NM_012751.1
Glucose-6-Phosphatase Catalytic Subunit 1	*G6pc1*	ACCCTGGTAGCCCTGTCTTT	GGGCTTTCTCTTCTGTGTCG	150	NM_013098.2
Lactate Dehydrogenase A	*Ldha*	GGT GGT TGA CAG TGC ATA CG	AGG ATA CAT GGG ACG CTG AG	186	NM_017025.1
MYC Proto-Oncogene, BHLH Transcription Factor	*Myc*	CGA GCT GAA GCG TAG CTT TT	CTC GCC GTT TCC TCA GTA AG	170	NM_012603.2
Glutaminase	*Gls*	CACACACACGGATTTCTTGG	GCCGAAGCTGACTTTGAAAC	194	NM_012569.2
Mitochondrial Pyruvate Carrier 1	*Mpc1*	ACTTTCGCCCTCTGTTGCTA	GCACTGTCCCTTTCAAGAGC	199	NM_133561.1
Carnitine Palmitoyltransferase 2	*Cpt2*	TCC TCG ATC AAG ATG GGA AC	GAT CCT TCA TCG GGA AGT CA	237	NM_012930.1
Glutamic-Oxaloacetic Transaminase 2	*Got2*	ACC ATC CAC TGC CGT CTT AC	TCT TGA AGG CTT CGG TCA CT	185	NM_013177.2
Peroxisome Proliferator Activated Receptor Alpha	*Ppara*	TCA CAC AAT GCA ATC CGT TT	GGC CTT GAC CTT GTT CAT GT	177	NM_013196.1
CD36 Molecule	*Cd36*	GCAACAACAAGGCCAGGTAT	AAGAGCTAGGCAGCATGGAA	155	NM_031561.2
Solute Carrier Family 27 Member 1	*Slc27a1*	CCTCACATCACAGCAGGAGA	GCTCTGTCCACACCCTTCAT	238	NM_053580.2
PPARG Coactivator 1 Alpha	*Ppargc1a*	ATGTGTCGCCTTCTTGCTCT	ATCTACTGCCTGGGGACCTT	180	NM_031347.1
Protein Kinase AMP-Activated Catalytic Subunit Alpha 2	*Prkaa2*	AGCTCGCAGTGGCTTATCAT	GGGGCTGTCTGCTATGAGAG	179	NM_023991.1
Acyl-CoA Dehydrogenase Medium Chain	*Acadm*	CAA GAG AGC CTG GGA ACT TG	CCC CAA AGA ATT TGC TTC AA	154	NM_016986.2
Ribosomal Protein L32	*Rpl32*	AGA TTC AAG GGC CAG ATC CT	CGA TGG CTT TTC GGT TCT TA	193	NM_013226
TATA Box Binding Protein	*Tbp*	CCT CTG AGA GCT CTG GGA TTG TA	GCC AAG ATT CAC GGT GGA TAC A	62	NM_001004198.1

## Data Availability

The data presented in this study are available upon request from the corresponding author.

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
