# Peer review of "The Lack of Synergy between Carvedilol and the Preventive Effect of Dexrazoxane in the Model of Chronic Anthracycline-Induced Cardiomyopathy"

_ijms, 2023, doi:10.3390/ijms241210202_

Round 1
Reviewer 1 Report
This article by Szponar et al investigates whether carvedilol exerts synergism with dexrazoxane to prevent doxorubicin-induced cardiomyopathy. The hypothesis is not covered in the literature and its clinical significance is introduced well.
Although results do not support the hypothesis, they can be valuable in the field. However, sole effects of carvedilol and dexrazoxane were not assessed in the experimental setup. Proofreading of the whole text and major revision of the Materials and Methods section are warranted.
Major comments:
1. Materials and Methods are not clear.
1.1. It is not clear how the total number of rats can be 60 if 5x10 rats were euthanized at week 11 and 5x10 rats at week 21.
1.2. Some sections are unnecessarily detailed, i.e., what an echocardiography is and how EIA works, but other parts are not explained in enough detail – i.e., experimental design.
1.3.Although included in the table, carvedilol-treated groups are not mentioned in the text of the Experimental design paragraph.
1.4. Similar to the histological assessment, an objective scoring system should be provided on how significance of clinical symptoms (ascites, enlarged liver) was decided.
2. It is questionable in Figure 2. why values are connected by lines. Even if the measurements were partly done on the same animals at week 11 and week 21 (which is not clear), it would be much more important to include individual data points in the figure, for example, in a bar graph form.
P- Please cite relevant articles that summarize the molecular mechanisms of doxorubicin induced cardiotoxicity. e.g.: PMID: 26386112
3
4. Additional control groups with carvedilol only and dexrazoxane only treatments should have been included in the experimental design to see individual effects of the applied agents in the experimental setup.
5. Abbreviations are not consistent. Carvedilol has three different abbreviations in the text (CVD, CRV, CVR) which is very confusing. Abbreviation RQ is not explained.
Minor comments:
1. A list of meanings of abbreviations could be included.
2. More background on how the genes for RT-PCR were selected could have been provided.
3. In the primer table, product size of Acadm is missing.
English language of the text should be proofread, as at some points, uncommon phrases are used or the context is difficult to understand (e.g., “rats were divided into two parts”, “material collection”, “acidophilus”, “the heart to do its substantial job”, “doxorubicin’s alone effects”).
Author Response
A point-by-point response to the Reviewers’ comments
We have read and carefully considered all the comments. We would like to thank the Reviewers for the constructive suggestions and remarks and we hope that our responses meet their expectations.
In the corrected version, we have highlighted (with ’tracked changes’) new sentences, rewritten parts, and some other changes. Some of these changes were made because of the comments of another reviewer.
Below we provide our point-by-point response to the Reviewer's comments and concerns.
Reviewer 1:
1: It is not clear how the total number of rats can be 60 if 5x10 rats were euthanized at week 11 and 5x10 rats at week 21.
Response 1: Thank you for noticing this typo, which is very important in meaning. The total number of rats at the beginning of the experiment was 100. Rats have been divided into two major groups for organizational reasons. For the first 11 weeks, experimental groups ‘I’ and ‘II’ were treated in the same way. However, groups ‘I’ (5x10 rats) were euthanized 11 weeks after the start of the experiment and the rats' samples were used for morphological, biochemical, and genetic investigations. Groups ‘II’ (5x10 rats) were euthanized 10 weeks later in the 21st week and the rats' samples were also used for morphological, biochemical, and genetic investigations. Moreover, in groups ‘II’ ECHO was performed in the 11th and 21st weeks. In addition to assessing general toxicity, all rats were under clinical observation, and liver macroscopic and ascites were evaluated. This information was inserted into the 4.2 section.
We understand that experimental design is complicated, so we agree with the reviewer that it should be as well described as possible. To facilitate understanding of the experimental model, we have placed Figure 1 at the beginning of the results, as the methodology according to the journal's requirements is under discussion. The first sentence of paragraph 4.1 and the 4.2 sections have been redrafted.
2: Some sections are unnecessarily detailed, i.e., what echocardiography is and how EIA works, but other parts are not explained in enough detail – i.e., experimental design.
Response 2: We agree with the Reviewer’s comment, unnecessary descriptions have been shortened (paragraphs 4.3 and 4.5). Section 4.2 has been supplemented with more detailed descriptions.
With regard to the changes presented above in point 1, information on the size of the groups was additionally provided for clarification purposes. All tests, i.e. ECHO, morphological, biochemical, and genetic, were performed on 10 animals in each experimental group, except for the DOX and DOX+CVD groups in the 21st week of the study, which were carried out on 5 and 6 animals, respectively.
3: Although included in the table, carvedilol-treated groups are not mentioned in the text of the Experimental design paragraph.
Response 3: We agree with the Reviewer. Therefore, according to the Reviewer's suggestion carvedilol-treated groups were described in the text of the Experimental design paragraph.
4: Similar to the histological assessment, an objective scoring system should be provided on how the significance of clinical symptoms (ascites, enlarged liver) was decided.
Response 4: Another form of table including an objective scoring system has been proposed
- It is questionable in Figure 2. why values are connected by lines. Even if the measurements were partly done on the same animals at week 11 and week 21 (which is not clear), it would be much more important to include individual data points in the figure, for example, in a bar graph form.
Response 5: We agree with the Reviewer’s suggestion. The previous graphs were illegible, and the data from the 'I' groups were unnecessary and could lead to misinterpretation. Therefore, we removed the cardiac ECHO data performed in 'I' groups. The new graphs show cardiac ECHO data performed in groups 'II', on the same 10 subjects (except for the DOXII and DOX+CVDII groups in the 21st week of the study, which were carried out on 5 and 6 animals, respectively) at 3-time points: before drug administration, one week after drug administration and 11 weeks after drug administration. We hope that the new graphs meet the expectations of the Reviewer.
- Please cite relevant articles that summarize the molecular mechanisms of doxorubicin-induced cardiotoxicity. e.g.: PMID: 26386112
Response 6: Indeed, the molecular mechanisms of DOX toxicity have not been adequately demonstrated. We have included citations in the introduction section.
- Additional control groups with carvedilol-only and dexrazoxane-only treatments should have been included in the experimental design to see individual effects of the applied agents in the experimental setup.
Response 7: The study did not include the carvedilol-only and dexrazoxane-only treatment groups. Our research plan included only those groups that were directly related to the research hypothesis, i.e. whether doxorubicin causes late cardiotoxicity, whether dexrazoxane has a protective effect on doxorubicin cardiotoxicity, and finally, whether carvedilol enhances the protective effect of dexrazoxane on doxorubicin cardiotoxicity. We agree with the Reviewer that including these groups would be valuable. However, the inclusion of carvedilol-only and dexrazoxane-only treatment groups was considered by the Ethics Committee and refused.
- Abbreviations are not consistent. Carvedilol has three different abbreviations in the text (CVD, CRV, CVR) which is very confusing. Abbreviation RQ is not explained.
Response 8: Abbreviations have been corrected and RQ has been explained.
- A list of meanings of abbreviations could be included.
Response 9: Thank you for this comment. A list of abbreviations used in the manuscript has been added under the materials and methods section.
- More background on how the genes for RT-PCR were selected could have been provided.
Response 10: The mRNA studies were used for the initial recognition of changes in gene expression of a given signaling pathway responsible for the transformation of energy metabolism. The genes for the study were selected arbitrarily. Many of them encode key proteins for energy metabolism pathways: glycolysis, gluconeogenesis, glutaminolysis, and lipid oxidation, as well as glucose and fatty acid transport proteins. We were guided by the importance of enzymes in a specific pathway when determining genes, and favoring those with irreversible reactions. In other cases, especially fatty acid transporters, the choice was difficult because the number of genes encoding different transporters is very large. For these processes, we selected genes whose role was described in the literature. A brief explanation has been added to the Materials and Methods section.
- In the primer table, product size of Acadm is missing.
Response 11: The product size of Acadm has been provided.
- English language of the text should be proofread, as at some points, uncommon phrases are used or the context is difficult to understand (e.g., “rats were divided into two parts”, “material collection”, “acidophilus”, “the heart to do its substantial job”, “doxorubicin’s alone effects”).
Response 12: The English language has been corrected by a native speaker and replaced with scientific language in places where uncommon phrases appeared. We will also use the MDPI service for minor English revision.
We thank the Reviewers for their thoroughness and the time to review this manuscript. We hope that thanks to their comments and suggestions, and our corrections, the manuscript will be able to be published soon.
Reviewer 2 Report
Reviewer comments and suggestions
The authors in this study evaluated anthracycline cardiotoxicity in rats treated with Carvedilol (CVD) in combination with Dexrazoxane (DEX). The studies were conducted using male Wistar rats receiving DOX (1.6 mg/kg b.w. i.p., cumulative dose: 16 mg/kg b.w.), DOX and DEX (25 mg/kg b.w. i.p.), DOX and CVD (1 mg/kg b.w. i.p.), or combined (DOX + DEX + CVD) for 10 weeks.
The result showed by the addition of CVD to DEX as a cardioprotective factor against DOX had no favorable advantages in terms of functional (ECHO), morphological (microscopic evaluation), and biochemical alterations (cTnI and BNP levels), as well as systemic toxicity (mortality and presence of ascites). From the experiment, the authors suggested that there is no justification to use simultaneous treatment of DEX and CVD in DOX-induced cardiotoxicity.
Overall, the manuscript needs a thorough revision based on the comments below.
- Line 37 Please add the full form of (cTnI and BNP levels)
- Line 85 Please explain the studies
- Line 88 Please check a typo error in this line
- Line 98 From which study the author's hypothesis their findings.. please explain here
- Table 1 The column should be arranged well
- Line 202-203 Please check the journal guideline whether it should be below the table
- Line 233-234 is there any possible for this
- Discussion first paragraph “Some inferences need to write only observation is not good to highlight”
- Please check the second paragraph of the discussion “Please check the numbering of references in this paragraph”
- Comments for discussion: It would be nice if the authors could add the table or figures number in the respective lines.
Author Response
A point-by-point response to the Reviewers’ comments
We have read and carefully considered all the comments. We would like to thank the Reviewers for the constructive suggestions and remarks and we hope that our responses meet their expectations.
In the corrected version, we have highlighted (with ’tracked changes’) new sentences, rewritten parts, and some other changes. Some of these changes were made because of the comments of another reviewer.
Below we provide our point-by-point response to the Reviewer's comments and concerns.
Reviewer 2
- Line 37 Please add the full form of (cTnI and BNP levels)
Response 1: Full forms of cTnI and BNP have been provided.
- Line 85 Please explain the studies
Response 2: We agree with the Reviewer’s remark. The mechanism was not precisely described, the data were provided.
- Line 88 Please check a typo error in this line
Response 3: We cannot find a typo, minor English revision will be performed by MDPI service.
- Line 98 From which study the author's hypothesis their findings.. please explain here
Response 4: The hypothesis is not covered in the literature which is a novel aspect of the study. Although results do not support the hypothesis, they can be valuable in the field. To date, there are no studies involving CVD and DEX combination for the prevention of DOX-induced cardiotoxicity.
- Table 1 The column should be arranged well
Response 5: The Table 1 layout has been improved and will be revised by the MDPI Editor.
- Line 202-203 Please check the journal guideline whether it should be below the table
Response 6: I did not find specific guidelines. According to the Editor's suggestions, I will correct descriptions in accordance with the requirements.
- Line 233-234 is there any possible for this
Response 7: Indeed, the Prkaa2 gene encodes Ampk, which has been shown to be downregulated as a response to DOX-induced cardiotoxicity, which is consistent with our study. Our results showed the AMPK-mediated cardioprotective effects of DEX and CVD pretreatment. However, due to the early nature of the data and studies that require confirmation, including Western blot, their relevance in this aspect has not been emphasized.
- Discussion first paragraph “Some inferences need to write only observation is not good to highlight”
Response 8: We agree with the reviewer's opinion, however, we ask that he be more specific.
- Please check the second paragraph of the discussion “Please check the numbering of references in this paragraph”
Response 9: The numbering of references has been checked – we cited the references used also in the introduction, which might be confusing.
- Comments for discussion: It would be nice if the authors could add the table or figure number in the respective lines.
Response 10: We thank the Reviewer for this comment, the tables and figures numbers have been added in the respective lines.
We thank the Reviewers for their thoroughness and the time to review this manuscript. We hope that thanks to their comments and suggestions, and our corrections, the manuscript will be able to be published soon.
Round 2
Reviewer 1 Report
All issues raised have been properly addressed.